# Induced cardiomyocyte maturation: Cardiac transcription factors are necessary but not sufficient

**Sophie Dal-Pra**[1,2], **Conrad P. Hodgkinson**[1,2], **Victor J. Dzau**[1,2]*

**1** Mandel Center for Hypertension and Atherosclerosis Research, Duke University, Durham, North Carolina, United States of America, **2** Cardiovascular Research Center, Duke University Medical Center, Durham, North Carolina, United States of America

* victor.dzau@duke.edu

**Data Availability Statement:** All relevant data are within the paper and its Supporting Information files.

**Funding:** The research conducted in these studies was supported by National Heart, Lung, and Blood

## Abstract

The process by which fibroblasts are directly reprogrammed into cardiomyocytes involves two stages; initiation and maturation. Initiation represents the initial expression of factors that induce fibroblasts to transdifferentiate into cardiomyocytes. Following initiation, the cell undergoes a period of maturation before becoming a mature cardiomyocyte. We wanted to understand the role of cardiac development transcription factors in the maturation process. We directly reprogram fibroblasts into cardiomyocytes by a combination of miRNAs (miR combo). The ability of miR combo to induce cardiomyocyte-specific genes in fibroblasts was lost following the knockdown of the cardiac transcription factors Gata4, Mef2C, Tbx5 and Hand2 (GMTH). To further clarify the role of GMTH in miR combo reprogramming we utilized a modified CRISPR-Cas9 approach to activate endogenous GMTH genes. Importantly, both miR combo and the modified CRISPR-Cas9 approach induced comparable levels of GMTH expression. While miR combo was able to reprogram fibroblasts into cardiomyocyte-like cells, the modified CRISPR-Cas9 approach could not. Indeed, we found that cardiomyocyte maturation only occurred with very high levels of GMT expression. Taken together, our data indicates that while endogenous cardiac transcription factors are insufficient to reprogram fibroblasts into mature cardiomyocytes, endogenous cardiac transcription factors are necessary for expression of maturation genes.

## Introduction

Several decades of research have given considerable insight into the early stages of heart development; however, the processes that drive maturation remain poorly understood. Cardiomyocytes, cardiac muscle cells which enable the heart to pump blood, initially develop from precursors expressing Mesp1 [1–5]. Mesp1 functions as the master regulator of cardiac development and is believed to control the expression, directly or indirectly, of the transcription factors Gata4, Hand2, Mef2C, Nkx2-5, and Tbx5 [4, 5]. These transcription factors form a cardiac transcription factor cascade that directs precursor cells towards a cardiomyocyte cell-fate [1, 2,

Institute, grant R01 HL131814-01A1 (https://www.nhlbi.nih.gov/). The funder had no role in study design, data collection and analysis, decision to publish, or preparation of the manuscript.

**Competing interests:** I have read the journal's policy and the authors of this manuscript have the following competing interests:Conrad Hodgkinson and Victor Dzau are co-founders of CV Gene Sciences. This company seeks to take miRNA based cardiac reprogramming to the clinic. This does not alter our adherence to PLOS ONE policies on sharing data and materials.

6]. Following commitment into the cardiomyocyte lineage, the committed cells develop into mature cardiomyocytes. During the period of maturation, specific ion channels are expressed; transverse tubules develop; intercalated discs connect adjacent cardiomyocytes to allow for simultaneous contraction; and sarcomeres appear and align[7]. While there has been significant progress in our understanding of cardiac development, the relationship between the cardiac transcription factor cascade and cardiomyocyte maturation remains uncertain.

Research into organ development gave rise to the idea of cellular reprogramming. This was demonstrated initially with MyoD; a transcription factor that was identified as an important regulator of muscle development. In these early cellular reprogramming studies, over-expression of MyoD was found to convert fibroblasts into muscle cells[8, 9]. Taking cues from cardiac development, several researchers demonstrated that the exogenous over-expression of components of the cardiac transcription factor cascade such as GMT (Gata4, Mef2C and Tbx5) and GMTH (GMT plus Hand2) directly reprogrammed fibroblasts into cardiomyocytes [10–12]. In an alternative approach, we utilized four microRNAs (miR-1, miR-133, miR-208, and miR-499) that are highly expressed in cardiomyocytes and conserved across species[13]. This combination of 4 microRNAs, which we call miR combo, directly reprogrammed fibroblasts into cardiomyocytes both in vitro[14–18] and in vivo[14, 19]. In comparison to cardiomyocyte generation via iPS cells; GMT/H and miR combo directly convert fibroblasts into cardiomyocytes without the need for an intermediate cell-type. Delivery of cardiac transcription factors or miR combo into the fibroblast initiates direct cardiac reprogramming. Following the initiation of direct cardiac reprogramming there is a period of maturation. Akin to cardiomyocyte development, the cell develops sarcomeres and acquires the electrophysiological properties of a mature cardiomyocyte. While there are a number of similarities between GMT/H and miR combo, both methods initiate cardiac reprogramming via epigenetic changes for example[17, 20], there is one notable difference: cardiac transcription factor expression levels. GMT/H cardiac reprogramming relies on the delivery of exogenous GMT/H genes into the fibroblast. In contrast, the constituent miRNAs of miR combo induce reprogramming by affecting the expression of endogenous genes. As one might expect, delivery of exogenous GMT/H genes into the fibroblast increases GMT/H expression levels by several orders of magnitude higher than miR combo. Considering the very different levels of cardiac transcription factor expression in both methods it is unclear what role endogenous cardiac transcription factors play in the development of cardiomyocytes. The role of cardiac transcription factors in cardiomyocyte development is further complicated by the recent finding that in vivo cardiac fibroblasts express GMT [21].

In this study, we wanted to understand the role of endogenous cardiac transcription factors in cardiomyocyte maturation. To that end we developed a modified Crispr approach that induced cardiac transcription factor expression to the same level as miR combo. While miR combo was able to reprogram fibroblasts into cardiomyocytes, the modified Crispr approach did not. Indeed, we found that cardiac reprogramming was relatively insensitive to cardiac transcription factor levels and only occurred with significant over-expression. Interestingly, the ability of miR combo to induce maturation was blocked by cardiac transcription factor knockdown suggesting that while cardiac transcription factors are necessary for maturation, by themselves they are not sufficient to induce maturation.

## Materials and methods

All experiments were carried out in accordance with all relevant Duke University guidelines and regulations.

## Animal experiments

Experiments using animals were approved by the Duke University Division of Laboratory Animals (DLAR) and the Duke Institutional Animal Care and Use Committee (IACUC). Protocol number A056-19-03

## Cardiac fibroblast isolation and culture

Mouse (C57BL/6) neonatal cardiac fibroblasts were isolated from 2 day old mouse neonates by incubating minced hearts in 0.1% collagenase-II for 30 minutes at 37˚C. Fibroblasts were isolated by centrifugation (400g for 5 minutes). Following isolation fibroblasts were cultured in growth media containing DMEM (ATCC, Catalogue number 30–2002) supplemented with 15%v/v FBS (Thermo Scientific Hyclone Fetal bovine serum, Catalogue number SH30071.03, Lot number AXK49952) and 1%v/v penicillin/streptomycin (Gibco, Catalogue number 15140–122, 100units Penicillin, 100ug/ml Streptomycin). Fibroblasts were passaged once the cells had reached 70–80% confluence using 0.05% w/v trypsin (Gibco, Catalogue number 25300–054). Freshly isolated fibroblasts were labelled as Passage 0. Experiments were conducted with cells at passage 2.

## Cardiac reprogramming with miR combo

Fibroblasts were seeded into 24 well plates at 9,000 cells per well. After 24 hours, the cells were transfected with transfection reagent alone (Dharmafect-I, ThermoScientific), with transfection reagent plus non-targeting microRNAs (negmiR), or with transfection reagent plus our previously reported combination of cardiac reprogramming microRNAs3 (miR combo, miR-1, miR-133, miR-208, miR-499) according to the manufacturer's instructions. After 24 hours the transfection complexes were removed and the cells incubated in DMEM supplemented with 15%v/v FBS. Media was changed every two or three days [22].

## qPCR

Total RNA was extracted using Quick-RNA MiniPrep Kit according to the manufacturer's instructions (Zymo Research). Total RNA (50ng-100ng) was converted to cDNA using a high capacity cDNA reverse transcription kit (Applied Biosystems). cDNA was used in a standard qPCR reaction involving FAM conjugated gene specific primers and TaqMan Gene Expression Master Mix (Applied Biosystems). Assay IDs can be found in[18]

## Activation of endogenous genes by dCas9-VPR

The dCas9-VPR (Addgene 63798), dCas9-HA (Addgene 47106), and dCas9-VPR-Ascl1 gRNA plasmids[23] were acquired from Dr. Gersbach. The guide-RNA (gRNA) plasmid, pSPgRNA [24], was acquired from Addgene (plasmid number 47108). Guide-RNAs were designed by the Weissman Lab[25]. Four to five gRNAs were designed for Gata4, Mef2C and Tbx5 (sequences below show gRNA and cloning site overhang).

Gata4 gRNAs:

#5 Top $^{5'}$caccgcatgcgcgcggaactctcg$^{3'}$; Bottom $^{5'}$aaaccgagagttccgcgcgcatgc$^{3'}$;

#6 Top $^{5'}$caccgctaagggagtcacgtgcaa$^{3'}$; Bottom $^{5'}$aaacttgcacgtgactccctagc$^{3'}$;

#7 Top $^{5'}$caccgcaagggccccgtagatctg$^{3'}$; Bottom $^{5'}$aaaccagatctacggggcccttgc$^{3'}$;

#9 Top ⁵′caccggacgtggaccactgagagt³′; Bottom ⁵′aaacactctcagtggtccac gtcc³′

Mef2C gRNAs:

#5 Top ⁵′caccgccgaagccgctggaagagg³′; Bottom ⁵′aaaccctcttccagcggctt cggc³′

#6 Top ⁵′caccggcaccgaagccgctggaag³′; Bottom ⁵′aaaccttccagcggcttcg gtgcc³′

#7 Top ⁵′caccgaggagaaagtggttgtctg³′; Bottom ⁵′aaaccagacaaccactttct cctc³′

#8 Top ⁵′caccggaagtgactggagacgaat³′; Bottom ⁵′aaacattcgtctccagtca cttcc³′

#9 Top ⁵′caccgactggagacgaatgggaaa³′; Bottom ⁵′aaactttcccattcgtctcc agtc³′

Tbx5 gRNAs:

#5 Top ⁵′caccgaacagccagcgagcagtgg³′; Bottom ⁵′aaacccactgctcgctggc tgttc³′

#6 Top ⁵′caccgactgagatcgtagggttta³′; Bottom ⁵′aaactaaaccctacgatct cagtc³′

#7 Top ⁵′caccgaagttgtcgggctccagaa³′; Bottom ⁵′aaacttctggagcccgacaa cttc³′

#8 Top ⁵′caccggggctccagaacggcttag³′; Bottom ⁵′aaacctaagccgttctgga gcccc³′

#9 Top ⁵′caccgggagacttgagagagactt³′; Bottom ⁵′aaacaagtctctctcaagt ctccc³′

Top and bottom strands for each gRNA (10µM final concentration) were annealed and phosphorylated with T4 Polynucleotide Kinase (New England Biolabs, MA) in T4 Polynucleotide kinase buffer. Phosphorylation and annealing was performed in a Bio-Rad iCycler thermocycler with the following parameters: 37˚C 30 minutes; 95˚C 5 minutes; ramp down to 25˚C at 5˚C minute⁻¹. Phosphorylated and annealed gRNAs were diluted 1:200 before use.

The pSPgRNA plasmid was digested with BbsI and purified with a Qiagen PCR clean-up™ kit according to the manufacturer's instructions. Once phosphorylated and annealed, gRNAs were cloned into the pSPgRNA plasmid (80ng) in a 20µl reaction containing digested pSPgRNA plasmid (80ng), gRNA (2µl), 10mM DTT (1µl), 10mM ATP (1µl), T4 DNA ligase (1µl; New England Biolabs), and water. The ligation was carried out for a minimum of two hours at 37˚C. The reaction (5µl) was then used to transform Stbl3 (50µl) bacteria (Thermo-Fisher) according to the manufacturer's protocol. Selection was performed with ampicillin. For each transformation, ten colonies were amplified overnight in LB Amp (LB broth containing 100µg/ml ampicillin) at 37˚C with constant agitation (225rpm). Following plasmid isolation (Qiaquick Miniprep kit, Qiagen, Hilden, Germany) a PCR reaction was performed to check for correct insertion of the gRNA. The PCR reaction contained the bottom strand of the gRNA as the forward primer (0.1µM) and a U6 primer (0.1µM) corresponding to the U6 site within the plasmid as the reverse primer. The ImmoMix Red Taq DNA polymerase was used according to the manufacturer's instructions (Bioline, Toronto, Canada). Once verified by sequencing gRNA plasmids were further amplified in LB Amp and endotoxin-free DNA isolated with an Zymo Plasmid Maxi-Prep kit (Zymo, CA).

Once confirmed, the gRNA plasmids were used to transfect MEFs or neonatal cardiac fibroblasts in conjunction with the dCas9-VPR plasmid. The dCas9-HA plasmid was used as a control. MEFs (5000 cells cm²) or neonatal cardiac fibroblasts (5000 cells cm²) were seeded twenty-four hours before transfection in growth media (DMEM (ATCC, VA) with 15% v/v

Fetal bovine serum (Gemini Bio-Products, CA) and 1xPenicillin-Streptomycin (Thermo-Fisher, MA)). On the day of transfection, two reaction tubes were set up. Transfections were carried out in either a 12-well or 24-well plate. For 1 well of a 12-well plate, Reaction tube 1 contained either the dCas9-HA or dCas9-VPR plasmid (250ng); gRNA plasmids (500ng total); PLUS Reagent (1.5μl, ThermoFisher, MA); and Optimem-media (to a total of 50μl, Thermo-Fisher, MA). For 1 well of a 12-well plate, Reaction tube 2 contained LTX Reagent (1.5μl, ThermoFisher, MA); and Optimem-media (to a total of 50μl, ThermoFisher, MA). After one 5 minute incubation at room temperature reaction tube 1 and reaction tube 2 were combined. Following a further 5 minute incubation at room temperature the complexes were added to the cells and media added (ATCC DMEM, 5% v/v FBS) added (250μl per well of a 12-well plate). Following incubation overnight, transfection complexes were removed and growth media added (neonatal cardiac fibroblasts: ATCC DMEM, 15% v/v FBS, 1xPencillin-Strepto-mycin; MEFs: DMEM High Glucose, 10%v/v NCS, 1xPenicillin-Streptomycin. Media was changed every two days for the duration of the experiment.

## Immunofluorescence & staining for sarcomeres

Cells were fixed with 2%v/v paraformaldehyde (EMS). Fixed cells were blocked in antibody buffer (5%w/v BSA, 0.1%v/v Tween-20, in PBS) for 1 hr at room temperature. Following blocking, cells were incubated overnight at 4oC with a-sarcomeric actinin antibody (Sigma, A7811, 1:100) in antibody buffer. After the overnight incubation, cells were washed three times in antibody buffer. Following washing, cells were incubated with Alexa-Fluor conjugated secondary antibodies (Invitrogen, Goat Anti-mouse 594nm) at a 1:500 dilution in antibody buffer for 1hr at room temperature. Nuclei were stained by DAPI at 1ug/ml for 30 minutes at room temperature in antibody buffer. Following washing in PBS to remove unbound complexes, immunofluorescence was measured using a Zeiss Axiovert 200 inverted microscope.

## GMT RNA expression

Gata4 (MR227022), Mef2C (MR207045) and Tbx5 (MR227369) expression plasmids were purchased from Origene, MD. Plasmid DNA was linearized with PmeI (New England Biolabs) and RNA generated via the mMessage mMachine T7 kit (ThermoFisher) according to the manufacturer's instructions. Neonatal cardiac fibroblasts, one day after seeding (4500 cells cm2) in growth media, were transfected with varying amounts of Gata4, Mef2C, and Tbx5 RNA to determine the optimal ratio for expression. Ratios of Gata4:Mef2C:Tbx5 employed were as follows: (1) 1:1:1 (G:M:T); (2) 0.5:1:0.5; (3) 0.25:1:0.25; (4) 0.125:1:0.125; (5) 0.063:1:0.063; (6) 0.038:1:0.038; and (7) 0:1:0. Optimal ratio was determined as that which induced the highest expression of maturation marker genes. Transfection (for ratio 1, 333ng of Gata4, Mef2C and Tbx5 were used per well of a 12-well plate) was accomplished with Lipo-MAX reagent (ThermoFisher). RNA was diluted in serum-free DMEM (ATCC) to a total volume of 25μl per well of a 12-well plate. To this was added 0.75μl LipoMAX which had been diluted in 24.25μl of serum-free DMEM (ATCC). After 20-minutes incubation at room temperature transfection complexes were added to the cells and growth media (250μl) added. Following overnight incubation, transfection complexes were removed and replaced with growth media. Media was changed every two days for the duration of the experiment.

## GMTH knockdown

siRNAs were purchased from Qiagen. In the initial screen, four siRNAs were used for Gata4, Hand2, Mef2C, and Tbx5. Please refer to[18] for the complete method. Analysis of gene expression was performed 4-days post-transfection. On the basis of highest knockdown

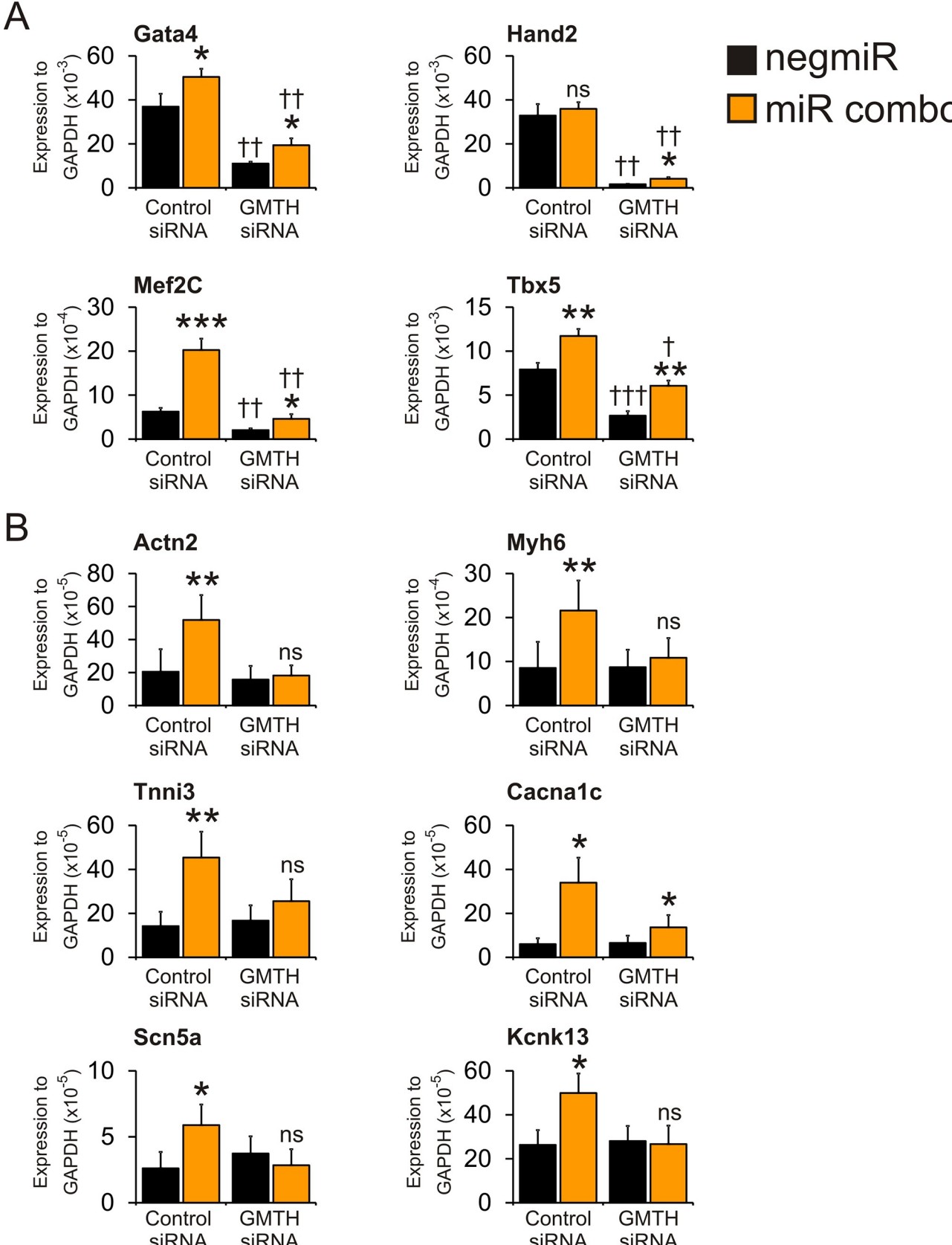

**Fig 1. GMTH Knockdown inhibits miR combo-mediated reprogramming.** Neonatal cardiac fibroblasts were transfected with negmiR (control) or miR combo and siRNAs that target Gata4, Tbx5, Mef2C, and Hand2 (non-targeting siRNA used as a control). **(A)** Expression of cardiac commitment genes was assessed four days after transfection by qPCR. N = 5. *Comparisons between negmiR and miR combo in the control siRNA and GMTH siRNA groups: *** $P < 0.0001$, ** $P < 0.001$, * $P < 0.05$. †Comparisons between GMTH siRNA groups and negmiR + control siRNA group: ††† $P < 0.001$, †† $P < 0.01$, † $P < 0.05$. **(B)** Expression of cardiomyocyte mature marker genes was assessed fourteen days after transfection by qPCR. N = 3–5. *Comparisons between negmiR and miR combo in the control siRNA and GMTH siRNA groups: ** $P < 0.001$, * $P < 0.05$, ns not significant.

efficiency one siRNA was chosen from the initial four tested (Gata4 siRNA #3 Cat no SI01009813; Mef2c siRNA #2 Cat no SI01303498; Tbx5 siRNA #5 Cat no SI02691738; Hand2 siRNA #1 Cat no SI01062775). These siRNAs were then used in conjunction with miRNA transfection. siRNA and miRNA transfection complexes were set-up independently as described above and added to the cells together. When siRNA and miRNA were used in conjunction the amount of complete media was reduced (250µl).

## Immunoblotting

Proteins were extracted in lysis buffer (62.5mM Tris pH7.4, 1% SDS, 1% protease inhibitor cocktail-I (Sigma)). Following extraction, proteins were subjected to SDS-PAGE and then transferred to nitrocellulose membranes (Bio-Rad). Membranes were blocked (5%w/v non-fat dry milk in TBS-Tween: 25mM Tris pH7.4; 137mM NaCl; 0.1%v/v Tween-20) and then incubated overnight at 4˚C with Mef2C (Cell Signaling, Cat no 5030S) or Gata4 (Cell Signaling, Cat no 36966S) antibodies diluted (1:1000 dilution) in TBS-Tween containing 5%w/v BSA. Following washing in TBS-Tween, membranes were incubated in HRP-conjugate secondary antibodies (Cell Signaling) diluted (1:1000 dilution) in TBS-Tween containing 5%w/v non-fat dry milk for 1 hour at room temperature. Following washing with TBS-Tween, membranes were developed with the ECL-Plus system according to the manufacturer's instructions (Amersham Biosciences). Bands were visualized with a G-Box (Syngene).

## Images

CorelDraw and Zeiss software (Axiovision Rel4.8 and Zen Blue) were used for the images in this study

## Statistics

T-Tests were used. Data is shown as Mean ± SEM. Significance was regarded as a P-value < 0.05.

## Data availability

The datasets generated during and/or analysed during the current study are available from the corresponding author on reasonable request.

## Results

### Cardiac transcription factor knockdown inhibits miRNA based cardiac reprogramming

Direct cardiac reprogramming has also been achieved via pharmacological agents[26, 27] and the over-expression of transcription factors[11, 12, 28]. Transcription factor-based direct cardiac reprogramming involves the over-expression of cardiac transcription factors Gata4 (G), Mef2C (M), Tbx5 (T) and Hand2 (H) [11, 12, 28]. In contrast, we have previously shown that a combination of miRNAs called miR combo directly reprograms fibroblasts into

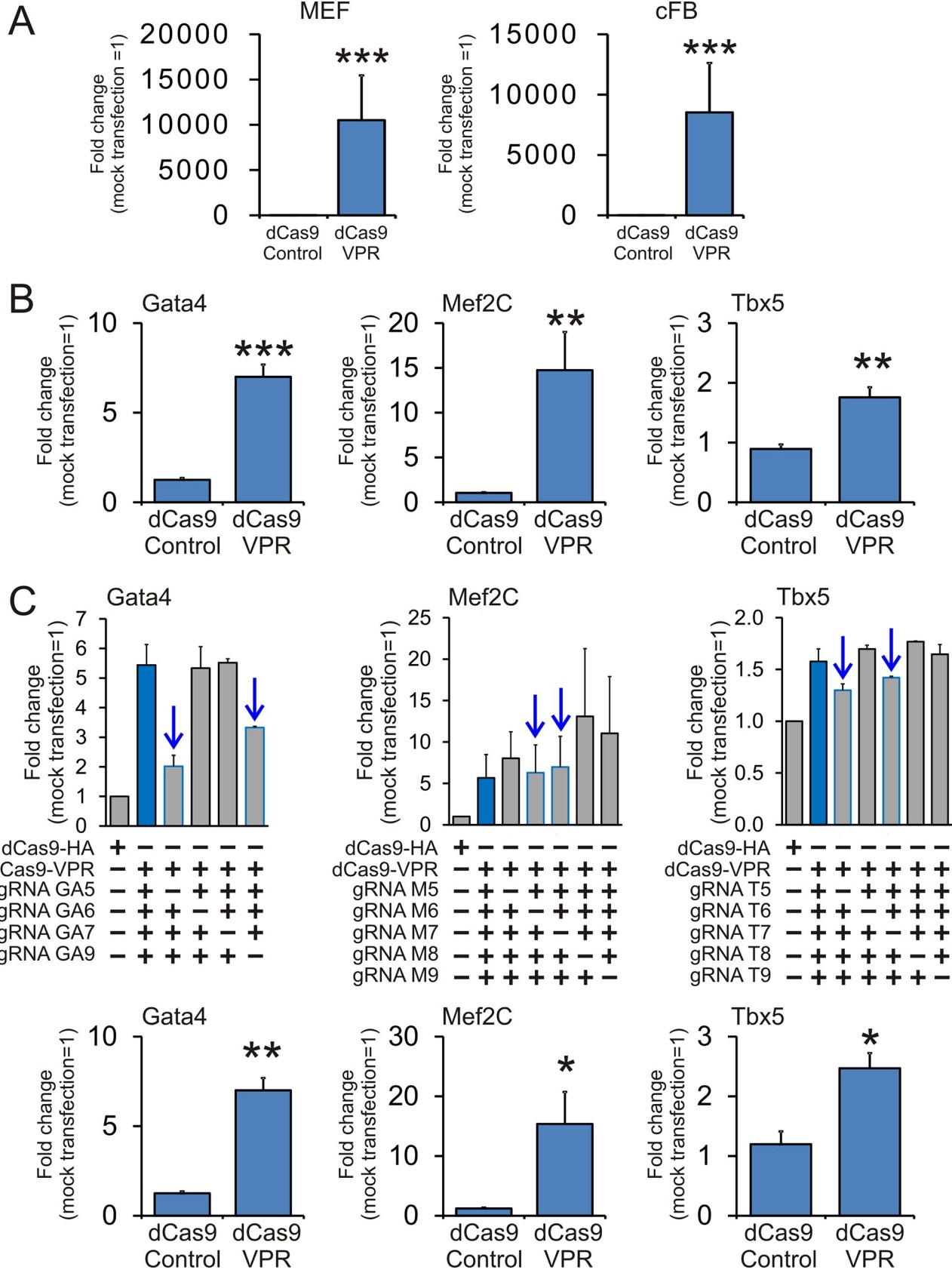

**Fig 2. Design and testing of the dCas9-VPR system for the activation of cardiac commitment factor genes. (A)** MEF (N = 5) and neonatal cardiac fibroblasts (N = 4) were transfected with dCas9-VPR and validated gRNAs for Ascl1. Expression was assessed three days after transfection by qPCR and shown as a fold change compared to a control plasmid. **(B)** MEF cells (N = 5) were transfected with dCas9-VPR and 4 gRNAs for Gata4 and 5 gRNAs for Mef2C and Tbx5. After 3 days, commitment marker expression was determined by qPCR and shown as a fold change compared to the mock transfection. **(C)** Top panels: MEF cells (N = 2) were transfected with dCas9-VPR and various combinations of the original pool of gRNAs. After 3 days, commitment marker expression was determined by qPCR and shown as a fold change compared to the mock transfection. The arrows indicate the active gRNAs in each pool; i.e. gRNAs which when removed from the pool of gRNAs, prevent dCas9-VPR from inducing gene expression. In each pool, two active gRNAs were identified and these two gRNAs were used for subsequent experiments. Bottom panels: MEF cells (N = 3) were transfected with the top two guide-RNAs for Gata4, Mef2C and Tbx5. After 3 days, commitment marker expression was determined by qPCR and shown as a fold change compared to the mock transfection.

cardiomyocytes[14–18, 22, 29]. While both GMT/GMTH over-expression and our miR combo reprogram fibroblasts into cardiomyocytes they do so despite very different levels of cardiac transcription factor expression. GMT/GMTH over-expression increases the expression of the GMTH factors by more than 10,000 fold; miR combo induces GMTH expression by 1.5 to 5 fold depending upon the cardiac transcription factor [14, 17].

In the first instance we wanted to determine if endogenous cardiac transcription factor expression was necessary for the induction of expression of maturation genes by miR combo. To that end, we used siRNAs to target the cardiac transcription factor cascade components Gata4, Mef2C, Tbx5 and Hand2 (GMTH). Knockdown efficiency was robust (Fig 1A): GMTH expression was significantly reduced in both control negmiR and miR combo transfected fibroblasts (Fig 1A). The expression level of GMTH had a significant impact on the ability of miR combo to induce the expression of mature cardiomyocyte markers. In control siRNA transfected cells, miR combo strongly induced mature cardiomyocyte marker expression (Fig 1B). However, in GMTH siRNA transfected cells, miR combo, despite inducing GMTH expression, was generally unable to induce mature cardiomyocyte marker expression (Fig 1B). Taken together, these results demonstrate the necessity of GMTH for miR combo reprogramming. Moreover, they also suggest that GMTH levels are an important determinant of their ability to induce maturation.

## dCas9-VPR and miR combo induce similar levels of cardiac transcription factor expression

To further investigate the role of endogenous GMT we induced expression of GMT by directly activating the endogenous GMT genes. Activation of the endogenous GMT genes was achieved via a modified CRISPR system. In this modified CRISPR system, the nuclease activity of the Cas9 is ablated through two point mutations and this nuclease-dead Cas9 (dCas9) is coupled to a gene activator. In our study, we chose to couple dCas9 to the gene activator VPR (dCas9-VPR)[23]. We first demonstrated that the dCas9-VPR system was able to induce gene expression in cardiac fibroblasts by using previously validated gRNAs to the Ascl1 gene[30]. As expected, Ascl1 expression was robust in both MEFs and cardiac fibroblasts (Fig 2A). Initially, we used four or five gRNAs from a published CRISPRa library[25] to activate the endogenous genes for Gata4, Mef2C and Tbx5 in MEFs (Fig 2B). We then wanted to determine which gRNAs in each pool were the active gRNAs. To do this we tested the effect of removing one gRNA from the pool. As shown in Fig 2C, the majority of gRNA activity was found with two gRNAs per gene (Fig 2C, marked with arrows). For example, transfecting fibroblasts with the dCas9-VPR construct and all four Gata4 targeting gRNAs, Gata4 expression significantly increased (Fig 2C). However, when gRNAs GA5 and GA9 were removed from the pool of Gata4 targeting gRNAs, there was no increase in Gata4 expression (Fig 2C).

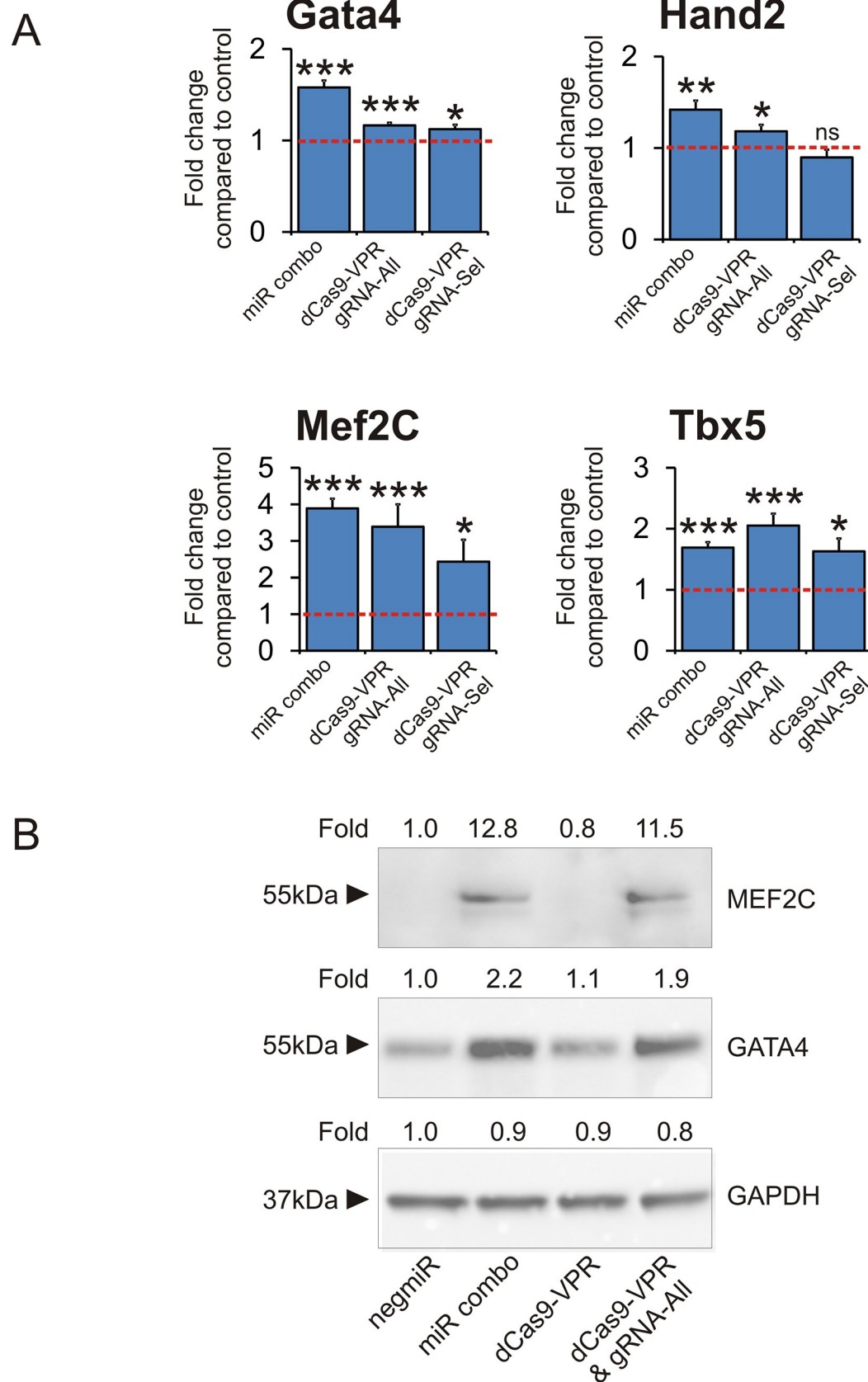

**Fig 3. dCas-VPR and miR combo have identical effects on cardiac transcription factor expression.** miR combo, dCas9-VPR + gRNA-All, or dCas9-VPR + gRNA-Sel, and their respective controls were used to transfect neonatal cardiac fibroblasts. Four days after transfection cardiac transcription factor mRNA (A) and protein (B) was assessed. **(A)** Expression of cardiac transcription factors was assessed four days after transfection by qPCR. N = 5. ***P<0.0001, **P<0.001, *P<0.05. **(B)** Protein levels of the cardiac transcription factors Mef2C and Gata4 were determined by immunoblotting. GAPDH was used as a loading control. Fold change compared to the negmiR control are shown for each immunoblot. N = 2. Representative cropped images are shown. Full-length blots are presented in S1 Fig.

Following these validation studies, cardiac fibroblasts were transfected with the dCas9-VPR as well as two distinct gRNA pools: gRNA-All which contained five gRNAs for each gene of GMT (15 gRNAs total); and gRNA-Sel which contained the two most potent gRNAs for each gene of GMT (6 gRNAs total). Of the two gRNA cocktails, gRNA-All was more potent than gRNA-Sel (Fig 3A). Importantly, miR combo and dCas9-VPR had a similar effect on the expression of endogenous cardiac transcription factors. Cardiac transcription factor mRNA levels were relatively similar between miR combo and dCas9-VPR (Fig 3A). Similarly, protein levels of the cardiac transcription factors Mef2C and Gata4 were virtually indistinguishable between miR combo and dCas9-VPR (Fig 3B). Protein measurements of Tbx5 were hampered by commercial antibody quality. Of the four cardiac transcription factors analyzed, Mef2C showed the highest level of induction with both miR combo and dCas9-VPR (Fig 3A and 3B). Recently, it has been demonstrated that high levels of Mef2C expression in comparison to Gata4 and Tbx5 are necessary for efficient reprogramming to cardiomyocytes[31].

## dCas9 mediated activation of cardiac transcription factor genes does not induce fibroblasts to reprogram into cardiomyocyte-like cells

Our miR combo has been demonstrated to reprogram fibroblasts into functional cardiomyocytes [14, 19]. Considering that dCas9-VPR and miR combo had similar effects on cardiac transcription levels we wondered if dCas9-VPR, like miR combo, was able to reprogram fibroblasts into cardiomyocyte-like cells. In agreement with our previous studies, miR combo induced cardiomyocyte-specific sarcomere and cardiomyocyte-specific ion channel gene expression in fibroblasts (Fig 4A). In contrast, activation of GMT via dCas9-VPR had no effect on either cardiomyocyte-specific sarcomere genes (Actn2, Myh6, Tnni3) or cardiomyocyte-specific ion channel expression (Scn5a, Cacna1c, Kcnk13, Kcnj2) (Fig 4A). Similarly, while miR combo was able to induce significant sarcomere formation, a signature of mature cardiomyocytes, activation of the endogenous GMT genes was only capable of producing immature cardiomyocytes with patchy and indistinct sarcomeres (Fig 4B with higher magnification images in Fig 4C and quantification in Fig 4D).

## Over-expression of cardiac transcription factors promotes cardiac reprogramming

These results demonstrate that endogenous cardiac transcription factors are necessary but insufficient for miR combo mediated maturation. However, it has been demonstrated that fibroblasts reprogram into mature cardiomyocytes when transfected with GMT factors[10–12] which gives rise to significant over-expression levels of GMT. As expected, introduction of GMT RNA into fibroblasts significantly increased GMT levels (Fig 5A). Significant over-expression of GMT, in agreement with previous studies, induced maturation marker expression between 4 and 20 fold (Fig 5B).

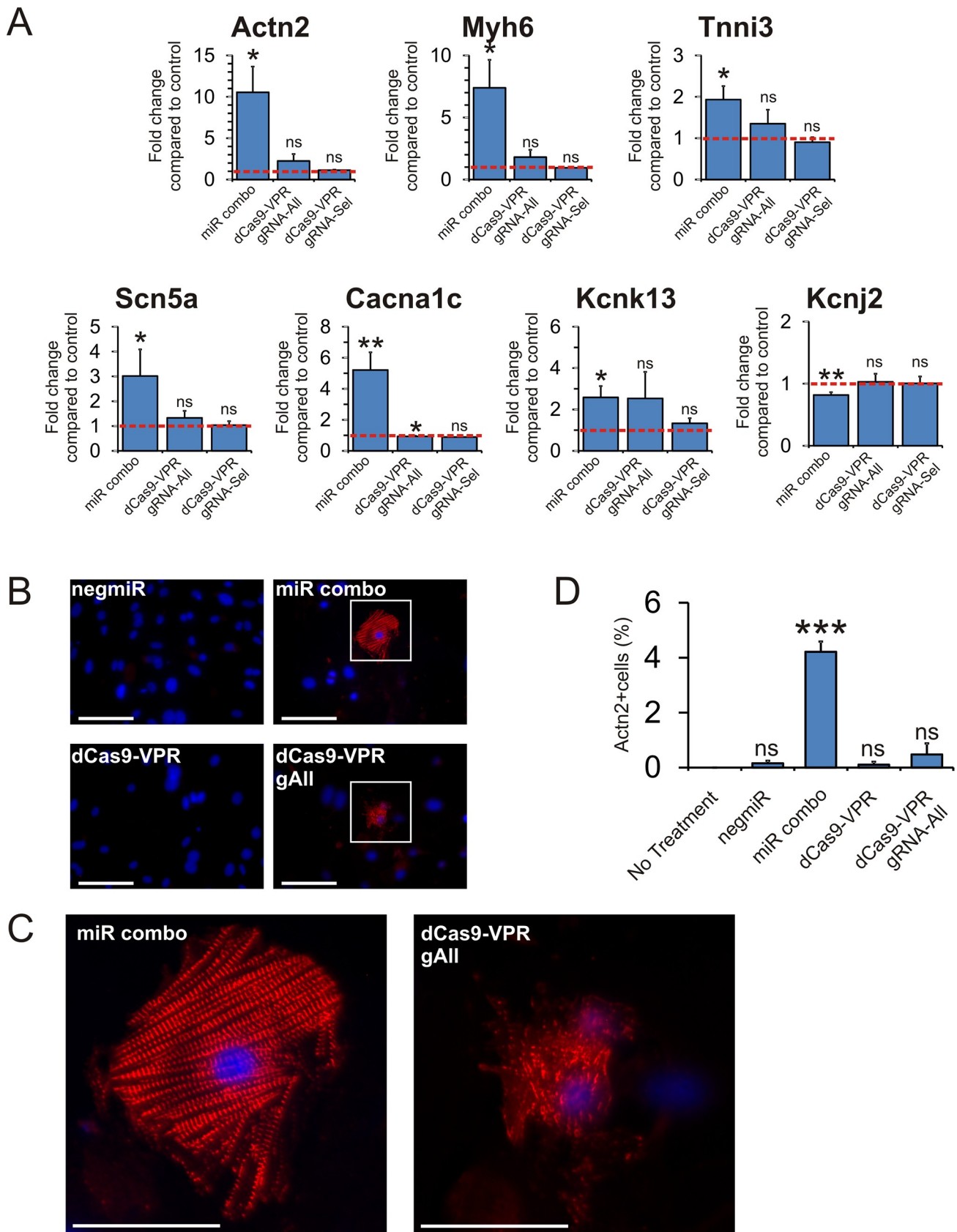

**Fig 4. Activation of cardiac transcription factor genes by dCas9-VPR does not sarcomere or cardiac ion channel expression in fibroblasts. (A)** Expression of cardiomyocyte mature marker genes was assessed fourteen days after transfection by qPCR. N = 5. *P<0.05, ns: not significant. **(B)** After 14 days culture in cardiomyogenic media, cardiomyocyte sarcomere formation was assessed by immunostaining for a-sarcomeric actinin (Actn2). Nuclei were stained with DAPI. Scale bar 50 microns. **(C)** Higher magnification images of panel B. Scale bar 25 microns. **(D)** Quantification of panel B. The total percentage of cells expressing Actn2 is shown. N = 3. ***P<0.001, ns not significant.

## Discussion

In this study, we show that endogenous cardiac transcription factors are necessary but not sufficient for cardiomyocyte maturation.

The expression of cardiac transcription factors, such as GMT, is an early event in the development of cardiomyocytes[1, 2, 6]. The role of cardiac transcription factors in the subsequent maturation of cardiomyocytes is less certain. Genetic ablation of components of the cardiac transcription factor cascade significantly impairs heart development[32–35]; demonstrating their importance. However, such studies do not demonstrate a role for cardiomyocyte maturation. Over-expression of components of cardiac transcription factor cascade, such as GMT and GMTH, induces fibroblasts to reprogram into cardiomyocytes[10–12]. Similarly, over-expression of Gata4 and Tbx5, two components of the cardiac transcription factor cascade, along with Baf60 induced cultured mouse mesoderm to differentiate into beating cardiomyocytes[36]. Moreover, over-expression of Gata4 and Nkx2-5, along with Serum Response Factor, significantly induced expression of the sarcomere component Actn2 in CV1 fibroblasts and Schneider 2 insect cells[37]. These and other studies[38–41] suggest that the cardiac transcription factor cascade is involved in cardiomyocyte maturation[42]. However, a common theme in these studies is over-expression. It is well known that over-expression can produce artefactual results[43–45]. Consequently, it remains unclear what role endogenous cardiac transcription factors play in cardiomyocyte maturation. Interestingly, we found that activation of the cardiac transcription factor cascade was unable to induce maturation. However, despite the cardiac transcription factor cascade being insufficient to induce maturation the cascade was still necessary as cardiac transcription factor knockdown inhibited cardiac reprogramming with respect to maturation gene expression. Taken together, these data suggest that rather than inducing maturation, endogenous cardiac transcription factors are needed to ensure that maturation genes can be induced. In further support of our study, we have recently shown that cardiomyocyte maturation can be achieved independently of any effect on cardiac transcription factor expression[18]. In this study, we showed that activation of the TLR3-NFκB pathway induced the expression of various sarcomere-related genes and significantly enhanced the maturation of reprogrammed fibroblasts into mature cardiomyocytes[18]. In contrast to the positive effects on maturation, activation of the TLR3-NFκB pathway had no effect on the cardiac transcription factor cascade [18]. The notion that initiation of cardiac reprogramming and cardiomyocyte maturation are distinct pathways is also supported by iPS studies. Many researchers have demonstrated that iPS cells can be robustly directed into the cardiomyocyte lineage[7, 46–48]. However, the resulting cardiomyocytes are usually immature[7, 46–48]. This lack of maturity impedes the clinical use of iPS derived cardiomyocytes. Unfortunately, improving the maturity of iPS derived cardiomyocytes has proven to be difficult[7, 46–48]. Our studies suggest that an emphasis on understanding the mechanisms of maturation will be fruitful in enhancing the maturity of iPS derived cardiomyocytes and realizing their significant clinical potential.

## Conclusions

In conclusion, we demonstrate that while endogenous cardiac transcription factors are necessary for cardiomyocyte maturation they are not sufficient.

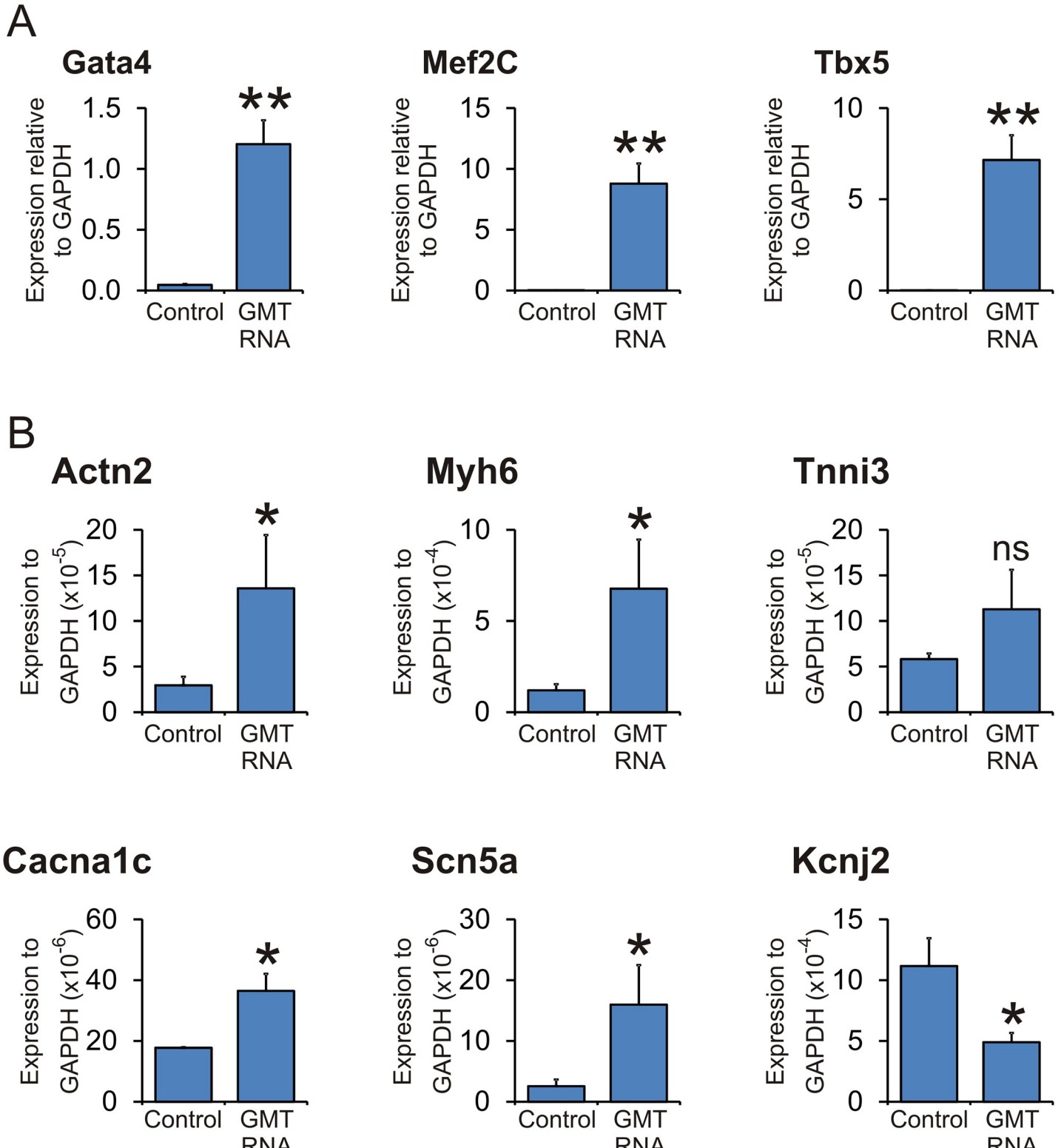

**Fig 5. Supra-physiological levels of GMT are necessary to induce cardiomyocyte maturation marker expression.** Neonatal cardiac fibroblasts were transfected with a combination of Gata4 RNA, Tbx5 RNA and Mef2C RNA (ratio empirically determined for maximal expression of maturation markers). **(A)** Expression of cardiac commitment genes was assessed four days after transfection by qPCR. N = 5. **(B)** Expression of cardiomyocyte mature marker genes was assessed fourteen days after transfection by qPCR. N = 5. *Comparisons were made between GMT RNA: **$P<0.001$, *$P<0.05$, ns not significant.

## Supporting information

**S1 Fig. Uncropped blots for Fig 3B.**
(TIF)

## Acknowledgments

Joshua Black, Charles Gersbach, Akshay Bareja and Alan Payne provided technical assistance.

## Author Contributions

**Conceptualization:** Sophie Dal-Pra, Conrad P. Hodgkinson, Victor J. Dzau.

**Data curation:** Sophie Dal-Pra, Conrad P. Hodgkinson.

**Formal analysis:** Sophie Dal-Pra, Conrad P. Hodgkinson, Victor J. Dzau.

**Funding acquisition:** Victor J. Dzau.

**Investigation:** Sophie Dal-Pra.

**Methodology:** Victor J. Dzau.

**Project administration:** Conrad P. Hodgkinson.

**Supervision:** Victor J. Dzau.

**Writing – original draft:** Sophie Dal-Pra, Conrad P. Hodgkinson, Victor J. Dzau.

**Writing – review & editing:** Conrad P. Hodgkinson.

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
