## [Decision Letter · Decision Letter 0]

25 Sep 2019

PONE-D-19-25053

Induced Cardiomyocyte maturation: cardiac transcription factors are necessary but not sufficient

PLOS ONE

Dear Dr. Hodgkinson,

Thank you for submitting your manuscript to PLOS ONE. After careful consideration, we feel that it has merit but does not fully meet PLOS ONE’s publication criteria as it currently stands. Therefore, we invite you to submit a revised version of the manuscript that addresses the points raised during the review process.

We would appreciate receiving your revised manuscript by Nov 09 2019 11:59PM. To enhance the reproducibility of your results, we recommend that if applicable you deposit your laboratory protocols in protocols.io, where a protocol can be assigned its own identifier (DOI) such that it can be cited independently in the future. For instructions see: http://journals.plos.org/plosone/s/submission-guidelines#loc-laboratory-protocols

We look forward to receiving your revised manuscript.

Kind regards,

Guo-Chang Fan, PhD

Academic Editor

PLOS ONE

Journal Requirements:

'I have read the journal's policy and the authors of this manuscript have the following competing interests:Conrad Hodgkinson and Victor Dzau are co-founders of CV Gene Sciences. This company seeks to take miRNA based cardiac reprogramming to the clinic.'

Additional Editor Comments (if provided):

Reviewers' comments:

Reviewer's Responses to Questions

**Comments to the Author**

1. Is the manuscript technically sound, and do the data support the conclusions?

Reviewer #1: Yes

Reviewer #2: Yes

2. Has the statistical analysis been performed appropriately and rigorously? 

Reviewer #1: Yes

Reviewer #2: Yes

3. Have the authors made all data underlying the findings in their manuscript fully available?

Reviewer #1: Yes

Reviewer #2: Yes

4. Is the manuscript presented in an intelligible fashion and written in standard English?

Reviewer #1: Yes

Reviewer #2: Yes

5. Review Comments to the Author

Reviewer #1: In this manuscript, Dal-Pra et al showed some interesting data that using dCas9-VPR to activate endogenous cardiac transcription factors GMTH (Gata4, Mef2C, Tbx5, Hand2) is not sufficient to reprogram fibroblast into cardiomyocyte like cells. However, miR combo is able to reprogram fibroblast into cardiomyocyte with similar expression of GMTH and the reprogramming was blocked by knocking down the endogenous expression of the GMTH. The experiments are well-designed and the data is convincing in general. Some minor points are as follows:

1 Fig.2C are not well-labelled. In the method, gRNA #5, #7, #8 and #9 are used, but #8 is missing in Gata4 panel in Fig.2C. In addition, the sentence in the main text “Further studies indicated that the majority of endogenous gene activation was found with two gRNAs per gene(Figure 2C, marked with arrows)” doesn’t match the figure legend – “one gRNA from the original pool of five was removed”. Please make the figure legend and the text consistent.

2 It is interesting that miR Combo is able to induce cardiomyocyte reprogramming while dCas9-VPR can’t. Given the comparable expression level of endogenous GMTH between these two condition, does this suggest other molecular mechanisms responsible for cardiomyocyte reprogramming induced by miR Combo?

Reviewer #2: This manuscript studies the function of cardiac development transcription factors in cardiomyocytes maturation. Authors used miR combo and the modified CRISPR-Cas9 methods induced cardiac transcription factors Gata4, Met2C, Tbx5, and Hand2 expression. The result showed that miR combo can reprogram fibroblasts into cardiomyocyte like cell, however, the modified CRISPR-Case9 approach can not induce cardiomyocytes maturation. These results suggested that endogenous cardiac transcript factors are critical factors for expression of maturation genes, but not sufficient for induction of maturation. This is very interesting study and this preliminary finding will be important for understanding mechanisms of cardiomyocytes maturation. The following are some comments on this manuscript.

1. In material and methods section, authors should briefly describe qPCR methods, cardiac fibroblast culture methods, cardiac reprogramming with miR combo and Western blotting methods.

2. In legend of Figure 2, authors wrote D section, but there is no D section. The expression of Ascl1 increased 10000 times in MEF and 7500 time in cFB compared to control, but the expression of Gata4, Mef2c, and Tbx5 increased 6 times, 15times,and 1.8 times compared to control. Why do activation of these transcription factors by dCas-VPR and gRNA were much lower than that of Ascl1?

3. The font of title in all figure should be consistent.

4. In Figure 3 B, antibody GATA4 and MEF2C were not listed in material and methods

5. In Figure 4B, the resolution of actinin staining imaging is too low, authors should provide high quality imaging. In legend of Figure 4B, authors should add scar bar size. The percentage of Actin2+ cells are too low. Whether staining methods cause this lower number Actin2+ cells? Also, antibody actinin was not listed in material and methods .

6. PLOS authors have the option to publish the peer review history of their article (what does this mean?). If published, this will include your full peer review and any attached files.

Reviewer #1: No

Reviewer #2: No

---

## [Author Response · Author response to Decision Letter 0]

27 Sep 2019

Response to reviewers

Journal Requirements:

• We have amended the manuscript to meet PLOS ONE requirements.

• We provide the uncropped blots in a Supplementary file. As requested we have also put this information into the Cover Letter.

'I have read the journal's policy and the authors of this manuscript have the following competing interests:Conrad Hodgkinson and Victor Dzau are co-founders of CV Gene Sciences. This company seeks to take miRNA based cardiac reprogramming to the clinic.'

• We confirm that our competing interest does not alter our adherence to PLOS ONE policies on sharing data and materials.

• As requested, we have updated the Competing Interests statement in the cover letter accordingly.

• We apologize for this oversight. The data in question is not a core part of the research presented in our study. Consequently, as suggested, we have removed the phrase that refers to the data not shown.

Review Comments to the Author

Reviewer #1: 

In this manuscript, Dal-Pra et al showed some interesting data that using dCas9-VPR to activate endogenous cardiac transcription factors GMTH (Gata4, Mef2C, Tbx5, Hand2) is not sufficient to reprogram fibroblast into cardiomyocyte like cells. However, miR combo is able to reprogram fibroblast into cardiomyocyte with similar expression of GMTH and the reprogramming was blocked by knocking down the endogenous expression of the GMTH. The experiments are well-designed and the data is convincing in general. Some minor points are as follows:

• We would like to thank the reviewer for their comments regarding our study.

1 Fig.2C are not well-labelled. In the method, gRNA #5, #7, #8 and #9 are used, but #8 is missing in Gata4 panel in Fig.2C. In addition, the sentence in the main text “Further studies indicated that the majority of endogenous gene activation was found with two gRNAs per gene(Figure 2C, marked with arrows)” doesn’t match the figure legend – “one gRNA from the original pool of five was removed”. Please make the figure legend and the text consistent.

• There were originally four gRNAs to Gata4 and we have amended the methods section accordingly.

• In Figure 2B we transfected the cells with the dCas9-VPR construct and a pool of 4 or 5 gRNAs per gene. In Figure 2C we wanted to identify which gRNAs in the pool were the active gRNAs. To that end, we removed one of the gRNAs from the pool and measured the effect on gene expression. We did this with all of the gRNAs in each pool. We found that for each gene, two of the gRNAs had the largest effect on gene expression. In other words, when these gRNAs were removed dCas9-VPR was no longer able to induce the expression of the targeted gene. These active gRNAs were marked with arrows. We agree that the text and figure was confusing. As such we have modified both text and figure accordingly.

2 It is interesting that miR Combo is able to induce cardiomyocyte reprogramming while dCas9-VPR can’t. Given the comparable expression level of endogenous GMTH between these two condition, does this suggest other molecular mechanisms responsible for cardiomyocyte reprogramming induced by miR Combo?

• This is an interesting question. We agree with the reviewer that the data in our study suggests that miR combo uses additional mechanisms beyond increasing GMT expression to induce cardiomyocyte reprogramming. One of these additional mechanisms may be the TLR3 NFkB pathway (Stem Cells. 2018 Aug;36(8):1198-1209) and we included a discussion on this topic in the Discussion section of the manuscript.

Reviewer #2: 

This manuscript studies the function of cardiac development transcription factors in cardiomyocytes maturation. Authors used miR combo and the modified CRISPR-Cas9 methods induced cardiac transcription factors Gata4, Met2C, Tbx5, and Hand2 expression. The result showed that miR combo can reprogram fibroblasts into cardiomyocyte like cell, however, the modified CRISPR-Case9 approach can not induce cardiomyocytes maturation. These results suggested that endogenous cardiac transcript factors are critical factors for expression of maturation genes, but not sufficient for induction of maturation. This is very interesting study and this preliminary finding will be important for understanding mechanisms of cardiomyocytes maturation. The following are some comments on this manuscript.

• We would like to thank the reviewer for their comments regarding our study.

1. In material and methods section, authors should briefly describe qPCR methods, cardiac fibroblast culture methods, cardiac reprogramming with miR combo and Western blotting methods.

• We have added brief methods for qPCR, cardiac fibroblast culture, cardiac reprogramming with miR combo and Western blotting (immunoblotting).

2. In legend of Figure 2, authors wrote D section, but there is no D section. The expression of Ascl1 increased 10000 times in MEF and 7500 time in cFB compared to control, but the expression of Gata4, Mef2c, and Tbx5 increased 6 times, 15times, and 1.8 times compared to control. Why do activation of these transcription factors by dCas-VPR and gRNA were much lower than that of Ascl1?

• We have found that baseline expression is a key predictor of the level of gene activation by dCas9-VPR. Genes that have a low baseline level of gene expression are strongly induced by dCas9-VPR. In contrast, genes which are expressed at a moderate or high level are weakly induced by dCas9-VPR. In cardiac fibroblasts, Ascl1 is weakly expressed (>400,000-fold lower than GAPDH). However, baseline Gata4, Mef2C and Tbx5 expression is much higher (2,000-20,000-fold lower than GAPDH). Interestingly, when the genes are induced by dCas9-VPR, the expression levels of Ascl1, Gata4, Mef2C, and Tbx5 are broadly similar to each other (~500-fold lower than GAPDH). We are not sure why the cell appears to have a threshold for endogenous gene activation via Crispr. However, it is certainly an interesting area for future studies. 

3. The font of title in all figure should be consistent.

• Where necessary we have modified the font to ensure consistency.

4. In Figure 3 B, antibody GATA4 and MEF2C were not listed in material and methods

• We have added this information to the materials and methods section (see Immunoblotting section).

5. In Figure 4B, the resolution of actinin staining imaging is too low, authors should provide high quality imaging. In legend of Figure 4B, authors should add scar bar size. The percentage of Actin2+ cells are too low. Whether staining methods cause this lower number Actin2+ cells? Also, antibody actinin was not listed in material and methods.

• We have provided higher magnification images in Figure 4C.

• Scale bars have been added to Figures 4B and 4C.

• We inadvertently used the wrong scale on the graph. The percentage of Actn2+ cells in the miR combo group is ~4% and not ~0.04% as was originally stated.

---

## [Editor Report · Decision Letter 1]

1 Oct 2019

Induced Cardiomyocyte maturation: cardiac transcription factors are necessary but not sufficient

PONE-D-19-25053R1

Dear Dr. Hodgkinson,

We are pleased to inform you that your manuscript has been judged scientifically suitable for publication and will be formally accepted for publication once it complies with all outstanding technical requirements.

With kind regards,

Guo-Chang Fan, PhD

Academic Editor

PLOS ONE
---

## [Editor Report · Acceptance letter]

7 Oct 2019

PONE-D-19-25053R1 

Induced Cardiomyocyte maturation: cardiac transcription factors are necessary but not sufficient 

Dear Dr. Hodgkinson:

I am pleased to inform you that your manuscript has been deemed suitable for publication in PLOS ONE. Congratulations! Your manuscript is now with our production department. 

With kind regards,

on behalf of

Dr. Guo-Chang Fan 

Academic Editor

PLOS ONE